# Development of a Low-Density DNA Microarray for Detecting Tick-Borne Bacterial and Piroplasmid Pathogens in African Cattle

**DOI:** 10.3390/tropicalmed4020064

**Published:** 2019-04-12

**Authors:** Babette Abanda, Archile Paguem, Mbunkah Daniel Achukwi, Alfons Renz, Albert Eisenbarth

**Affiliations:** 1Institute of Evolution and Ecology, Department of Comparative Zoology, University of Tübingen, Auf der Morgenstelle 28, D-72076 Tübingen, Germany; achillepaguem@yahoo.fr (A.P.); alfons.renz@uni-tuebingen.de (A.R.); albert.eisenbarth@fli.de (A.E.); 2Programme Onchocercoses field station of the University of Tübingen, P.O. Box. 65 Ngaoundéré, Cameroon; 3Department of Biological Sciences, University of Ngaoundéré, P.O. Box 454 Ngaoundéré, Cameroon; 4Trypanosomosis Onchocerciasis Zoonoses Association for Research & Development, Research Laboratory, Bambili-Tubah, P.O. Box 59 Bamenda, Cameroon; mbunkahachukwi@gmail.com

**Keywords:** tick-borne pathogen, low-cost and low-density-array, Reverse Line Blot, *Anaplasma*, *Ehrlichia*, *Rickettsia*, *Theileria*

## Abstract

In Africa, pathogens transmitted by ticks are of major concern in livestock production and human health. Despite noticeable improvements particularly of molecular screening methods, their widespread availability and the detection of multiple infections remain challenging. Hence, we developed a universally accessible and robust tool for the detection of bacterial pathogens and piroplasmid parasites of cattle. A low-cost and low-density chip DNA microarray kit (LCD-Array) was designed and tested towards its specificity and sensitivity for five genera causing tick-borne diseases. The blood samples used for this study were collected from cattle in Northern Cameroon. Altogether, 12 species of the genera *Anaplasma*, *Ehrlichia*, *Rickettsia* and *Theileria*, and their corresponding genus-wide probes including *Babesia* were tested on a single LCD-Array. The detection limit of plasmid controls by PCR ranged from 1 to 75 copies per µL depending on the species. All sequenced species hybridized on the LCD-Array. As expected, PCR, agarose gel electrophoresis and Sanger sequencing found significantly less pathogens than the LCD-Array (*p* < 0.001). *Theileria* and *Rickettsia* had lower detection limits than *Anaplasma* and *Ehrlichia.* The parallel identification of some of the most detrimental tick-borne pathogens of livestock, and the possible implementation in small molecular-diagnostic laboratories with limited capacities makes the LCD-Array an appealing asset.

## 1. Introduction

Tick-borne pathogens (TBP) are of high veterinary and medical importance worldwide. To evaluate the risk of exposure of TBPs in a livestock or human population, effective surveillance and monitoring practices are needed. For cattle and other livestock, the published literature highlights the importance of protozoa of the genera *Babesia* and *Theileria*, bacteria of the genera *Anaplasma*, *Ehrlichia* and *Rickettsia*, and arboviruses as etiologic agents of many diseases, of which a number of them have zoonotic potential [1]. Especially in developing countries, routine diagnostic approaches for the identification of TBPs are generally based on microscopic examination of blood smears [2,3] or serological assays [4,5]. While those techniques require only moderate investments for equipment and infrastructure, they have limitations regarding specificity and sensitivity (microscopy) [6,7,8], or tend to cross-react with closely related species (enzyme-linked immunosorbent assays) [9]. Furthermore, commercially available kits of the former are often not financially affordable for veterinary laboratories in low income endemic countries. Molecular tools based on PCR [10] and nowadays NGS are becoming more widespread, with NGS being economically viable when used for large sample sizes [11].

The DNA microarray technology of PCR-amplified products combines high throughput, sensitivity, specificity and reproducibility [12]. Its function is based on the reverse line blot (RLB), in which specific oligonucleotide spots (probes) are immobilized on a solid surface (Figure 1). When a target sample with complementary DNA sequence is added, it hybridizes with the probe where it is detected by a fluorescent, chemiluminescent or biotinylated label. The synchronous detection of a multitude of species in the same genetic material has contributed to its popularity in infectious disease diagnostics [10,13]. Low-density DNA microarrays such as the LCD-Array are designed to detect much lower numbers of pathogenic agents than high-density microarrays [14]. However, they are optimized for minimal input of equipment, workflow, costs and expenditure of time, and therefore suitable for small diagnostic laboratories in low and middle income developing countries [14,15].

In TBP epizootiology, the mostly used RLB application has been a mini-blotter coupled with a membrane where the probes of interest have been priorly linked to [10,13]. Although any desirable probes can be attached to the membrane prior to testing, the setup necessitates a high skill level in handling and optimization. Hence, for routine TBP identification a “ready to use” array or biochip for low to medium sample numbers with standardized protocol and reagents would be highly desirable.

In this paper we describe the development and testing of a novel LCD microarray for TBP, based on an already established biochip platform from a commercial provider (Chipron, Berlin, Germany). The same platform has been adapted for the detection of human mycobacteria [16], viruses [14,17], fungi [18] and in food safety [12]. In the field of TBP, this array has been tested once for the two piroplasmidae genera *Babesia* and *Theileria* [19]. In our study, the PCR and LCD-Array also detect ribosomal RNA fragments (18S) of the genera *Babesia* and *Theileria*, and additionally bacterial 16S fragments of the genera *Anaplasma*, *Ehrlichia* and *Rickettsia*. The array design, protocol specifications and performance in comparison to PCR with Sanger sequencing are described and tested on a naturally exposed cattle population from North Cameroon.

## 2. Materials and Methods 

### 2.1. Sample Origin, DNA Extraction, PCR and Sanger Sequencing

The tested blood samples (*n* = 31) were collected from cattle in Northern Cameroon. Blood samples (5 mL in EDTA tubes) were taken from the jugular vein of animals and tested by PCR and agarose gel electrophoresis. Briefly, blood samples were centrifuged at 3000 rpm using the Z380 laboratory centrifuge (Hermle Labortechnik, Wehingen, Germany) for 15 min and 300 µL of the erythrocyte and buffy coat was used for DNA extraction according to the manufacturer’s instructions of the Wizard Genomic DNA Purification Kit (Promega, Madison, WI, USA). Published primer pairs were used for the identification of the genera *Babesia*/*Theileria* [20] and *Rickettsia* [10]. Based on sequence alignments of the target species and ribosomal regions in GenBank, a new primer pair was designed for the detection of *Anaplasma*/*Ehrlichia*. The primer sequences and corresponding annealing temperatures are given in Table 1. To identify TBP-positive samples, a PCR reaction was done in 25 µL total volume combined as followed: 12.5 µL of the 2× RedMaster Mix (Genaxxon BioScience, Ulm, Germany) or 1 mM MgCl_2_, 0.5 mM 5× buffer, 200 µM nucleotides mix and 1 U GoTaq DNA polymerase (Promega, Madison, WI, USA). To the master mix, 10 pmol of each primer was added per reaction. One microliter of template DNA was added to 24 µL of mastermix reagents, and HPLC-grade water (Sigma Aldrich, Taufkirchen, Germany) was used as PCR negative control. Temperature cycles were programmed on a MasterCycler EPS 96-well thermocycler (Eppendorf, Hamburg, Germany): initial denaturation at 95 °C for 3 min, 35 cycles of 95 °C for 30 s, annealing temperatures (Table 1) for 30 s, 72 °C for 30 s, followed by a final elongation step of 72 °C for 10 min. Five microliter of the amplified products with 1 µL of loading buffer (Genaxxon BioScience, Ulm, Germany) were loaded on a 1.5% agarose gel with Tris Borate EDTA buffer (TBE) stained with Midori Green (Nippon Genetics Europe, Düren, Germany), run for about 40 min at 100 V, and photographed under UV light. The selected specimens with visible PCR product in the gel were prepared and submitted for DNA sequencing according to the provider’s recommendation (Macrogen Europe, Amsterdam, Netherlands). The retrieved sequence data was edited manually, MUSCLE aligned and analyzed with Geneious v9.1 (Biomatters, Auckland, New Zealand) and the GenBank nucleotide database (National Center of Biotechnology Information, Bethesda, MD, USA).

### 2.2. LCD-Array Specification and Validation

To allow the detection on the array, a similar PCR reaction was done with one of the paired primers being biotinylated at the 5′-end (Table 1) at a concentration 10-times higher than the corresponding non-biotinylated primer. Moreover, 10 more temperature cycles were added to increase template amplification for hybridization. For sensitivity tests, twelve constructs on the plasmid vector pUC57 (Baseclear, Leiden, Netherlands) with inserts of the following gene loci and species were used as positive controls: For 16S rRNA *Anaplasma centrale*, *A. marginale*, *A. platys* (*A. sp.* ‘Ommatjenne’), *A. sp.* ‘Hadesa’, *E. canis*, *Ehrlichia ruminantium*, *Rickettsia africae* and *R. felis*. For 18S rRNA *Theileria annulata*, *T. mutans*, *T. parva* and *T. velifera* was used. The concentration of plasmid constructs was measured by the Qubit 4 Fluorometer (Thermo Fisher Scientific, Waltham, MA, USA), and the number of copies calculated from the amount of DNA in ng and the length of the template in base pairs using the formulae described on the webpage http://cels.uri.edu/gsc/cndna.html (URI Genomics and Sequencing Center). Ten-fold serial dilutions in HPLC-grade water (Sigma Aldrich, Taufkirchen, Germany) as solvent were prepared and used as PCR templates, resulting in target concentrations ranging from 1 to 75 plasmid copies per reaction. Those dilutions of plasmids were amplified by PCR and loaded on gel electrophoresis, as well as tested on the LCD-Array using the first dilution with no detectable PCR product in the agarose gel, respectively for each of the species amplicons.

The LCD-Array consists of a transparent, pre-structured polymer support, with 50 by 50 mm dimensions. Each array had eight individually addressable sample wells where the probes are spotted on the surface as 19 to 28-meres of oligonucleotides using contact-free piezo dispensing technology [14]. The array presently used contained 33 probe spots of which three are proprietary kit controls (‘hybridization controls’), and 30 genera- or species-specific probes in duplicates as controls in case of mechanical failure (Figure 1). Altogether, 12 TBP species and 3 genera or groups of genera (“catch all”) were included. The probes were selected according to highest genus or species coverage in GenBank. Parameters of selection were the exclusion of unintended hybridization with other genera or species, melting temperature optimum for the LCD-Array, and distance of the hybridization site to the biotinylated primer.

### 2.3. LCD-Array Workflow

Single amplicons produced by each of the generic primer pairs or mixtures of the three species groups—each containing one biotinylated primer—were added at a final volume of 10 µL (for single product) and in equal proportions (3.3 µL for the mixture) to the LCD-Array according to the manufacturer’s protocol (Chipron, Berlin, Germany). Briefly, 10 µL of the mixture was added to 24 µL Hybridization Mix (Chipron), and 28 µL thereof was applied per sample well. The chip was placed in the kit’s humidity chamber and incubated in a 35 °C water bath for 30 min. Afterwards, washing steps were conducted with the supplied washing buffer for about 2 min successively in three small tanks filled with about 200 mL of 1× washing buffer. The slide was dried by spinning in the Chip-Spin centrifuge (Chipron, Berlin, Germany) for 15 s. Then, 28 µL of the previously combined horseradish peroxidase—streptavidin conjugate (Chipron) was added to the array for labeling, and incubated for 5 min. Subsequently, the array was washed and dried as previously indicated. Finally, 28 µL of the staining solution tetra methyl benzidine was added to each sample well. After 5 min incubation at room temperature, the staining process was stopped by washing once for 10 s and drying as described before. All tanks were filled with new washing buffer after each step. The LCD-Array was analyzed using the SlideScanner PF725u with the software package SlideReader V12 (Chipron, Berlin, Germany) for automated identification. By default, the cut-off value for positive detection was 2000 pixel values.

To test the specificity and the sensitivity of the assay, 10 µL of the PCR amplification products of each recombinant positive control plasmid was submitted to the array. The template concentrations were one order below the limit of detection by agarose gel electrophoresis as described above. For cross hybridization tests, PCR products of all three genera/groups of genera were mixed at equal volume. Cattle field samples (*n* = 31) were PCR amplified and tested on the LCD-Array for analogy with previously obtained sequencing results.

The statistical analysis was done using R v.3.4.2 (www.R-project.org). Data produced from both tests (sequencing and LCD-Array chip) were considered as paired data. The paired *t*-test was used to assess the difference between both diagnostics. A statistical p-value below 0.05 was considered significant.

## 3. Results

### 3.1. LCD-Array Performance of Synthetic Inserts (Plasmids)

All twelve plasmid constructs hybridized only with their respective probes, including “catch all” on the LCD-Array (Figure 2). The tested concentration of plasmid template on the array was 10 to 1000 times lower than on agarose gel (Table 2). Onagarose gel electrophoresis the product was still visible at 10^−8^ dilution for Theileria and Rickettsia, and for dilutions between 10^−5^ and 10^−7^ for Anaplasma and Ehrlichia (Figure 3).

### 3.2. LCD-Array Performance of Cattle Blood Samples from North Cameroon

All pathogens identified by Sanger sequencing in the field-collected blood samples were also detected on the LCD-Array. Furthermore, the array revealed co-infections of more TBPs which were not detected by the sequencing (Figure 4). Statistical comparison showed significant lower detection rates by sequencing as compared to the LCD-Array.

#### 3.2.1. *Anaplasma*

Of the 31 blood samples tested, *A. marginale* was detected in 61.3% (19/31), followed by *A. platys* 41.9% (13/31), *A. sp.* ‘Hadesa’ 41.9% (13/31), and *A. centrale* 41.9% (13/31). Sanger sequencing had consistently lower detection rates of 12.9%, 29.0%, 6.5% and 12.9% for the same species, respectively. In 26 of 29 positive cases (89.7%) both the species-specific and genus specific (“catch all”) probes were hybridizing. The remaining 3 of 29 positive cases reacted only with the *Anaplasma*/*Ehrlichia* “catch all” probe. From the 31 screened samples, 12 from the *Anaplasma*/*Ehrlichia* could not be sequenced. Of those unsuccessfully sequenced samples the LCD-Array identified 8 species.

#### 3.2.2. *Ehrlichia*

*Ehrlichia* species were detected in 17 (54.8%, 17/31) of the screened samples being significantly higher (*p* < 0.001) than the prevalence detected by Sanger sequencing (3.2%, 1/31). Among the unsuccessfully sequenced samples screened under the LCD-Array, *E. ruminantium* was found in co-infection with *A. centrale* and *A. marginale*. In another case *E. ruminantium* was found in co-infection with *A. marginale*. *E. canis* was found by sequencing and hybridized by its specific probe on the array in only one sample, however below the threshold of 2000 pixel values. From the 17 positive cases for *E. ruminantium*, 16 were also positive for the “catch all”. From the 31 screened samples, 12 from the *Anaplasma*/*Ehrlichia* primers could not be sequenced. The LCD-Array detected 8 of those samples being positive for *A. marginale* (*n* = 3), *E. ruminantium* (*n* = 3) and each co-infected specimens of *A. sp.* ‘Hadesa’, *A. marginale* and *A. platys*; *A. centrale*, *A. marginale* and *E. ruminantium,* and *A. marginale* and *E. ruminantium*.

#### 3.2.3. *Rickettsia*

*Rickettsia africae* and *R. felis* were detected on the LCD-Array in 16/31 (51.6%) and 4/31 (12.9%) of cases, respectively, being higher than the detection rates by Sanger sequencing 8/31 (25.8%) and 1/31 (3.2%) of cases, respectively. Eighteen of 20 cases positive for *Rickettsia* species (90%) were also hybridizing with the *Rickettsia*-“catch all” probe. The other two out of 20 samples (10%) were only positive for *Rickettsia* “catch all”. PCR amplicons identified by sequencing as bacteria related to *Klebsiella* or *Brevundimonas* did not hybridize with any probe on the LCD-Array. From the 21 PCR-positive samples with negative sequencing results 8 *R. africae* were detected by the microarray, 3 co-infected with *R. africae* and *R. felis*, and one with *R. felis*.

#### 3.2.4. *Babesia*

None of the samples was positively tested and confirmed for *Babesia spp*. Hence, the present LCD-Array did not include probes specific to *Babesia*. However, the *Babesia*/*Theileria* “catch all” probe is complementary to the 18S loci of the bulk of *Babesia spp*.

#### 3.2.5. *Theileria*

In accordance with the sequencing results, *Theileria mutans* and *T. velifera* were detected in high numbers (90.3%, 28/31, and 77.4%, 24/31, respectively). Detection by sequencing produced unknown *Theileria sp.* in 3 cases, *T. velifera* in one case, *T. mutans* in 17 cases, and *T. mutans* co-infected with *T. velifera* in 3 cases. In 85.7% (24/28) of the cases, *T. mutans* was found in co-infection with *T. velifera* which is significantly higher than recorded by Sanger sequencing of the PCR-product (13.6%; 3/22; *p* < 0.001). 26 of 28 positive animals (92.8%) were also signaling by the “catch all” probe. Both *T. annulata* and *T. parva* were not found neither by sequencing nor by LCD-Array. All PCR-positive samples with no outcome by sequencing (*n* = 5) were identified with the LCD-Array as *T. mutans* and co-infected with *T. velifera* (*n* = 3) and without (*n* = 2).

## 4. Discussion

The current LCD-Array based on the RLB method has been developed and used to test samples collected from cattle in the northern part of Cameroon. These samples have previously been screened for TBPs using conventional PCR and Sanger sequencing, and a subset of these samples is now being tested by the novel LCD-Array. Co-infection with up to six TBP per animal was common [20], yet difficult to detect by PCR and sequencing alone [13]. In such a scenario, utilization of generic primers poses the problem of correct allocation to the respective species or species complex. DNA sequencing without prior cloning of the less prevalent amplicons is often unsuccessful or distorts the whole readout making it at times incomprehensible [21]. Furthermore, the pathogen concentration in the host blood varies dramatically depending on the animal’s state of infection, making the identification challenging when present in very low concentrations. For *Theileria spp.* it is known that carrier animals persist with a low number of infected erythrocytes [22]. Moreover, competition for multiple PCR templates are further limiting factors for the detection of pathogens in low concentrations. In this study, the sensitivity tested on the LCD-Array was between 10 and 1000 times higher than by PCR and Sanger sequencing (Table 2).

The hybridization in some cases of only the “catch all” probe (Figure 4C for *Rickettsia*) suggests the presence of bacteria or parasite species not addressed by the LCD-Array. If DNA sequencing of the PCR product cannot unveil the species responsible for the hybridization, alternative gene loci generally used for molecular taxonomy (e.g., *cox*-I, GAPDH, etc.) could pave the way. The highly pathogenic piroplasmids *T. annulata* and *T. parva* were not confirmed in the blood samples, although three samples reacted with the corresponding hybridization spots below the cut-off value. Attempts to sequence those inconclusive specimens using primer pairs of species-specific target regions did not bring light to the effective presence of those pathogens. So far, outbreaks with high fatalities are only known in East Africa for *T. parva*, and North Africa for *T. annulata* [23]. By Sanger sequencing of the positively tested animals only *Theileria* species of low pathogenicity were discovered.

Specific probes for the genus *Babesia* were not included in the array because their presence could not be confirmed by PCR in our dataset. Previous infections of *Babesia* spp. may not be detectable by molecular tools as the pathogen can be completely cleared from the blood stream and even from organs [24]. The evidence of *Babesia* in a study from Northern Cameroon [2] could indicate current or very recent infection event in the sampled individuals, allowing its identification on Giemsa stained blood smears.

Reportedly more reliable than the real-time PCR for the detection of new pathogen strains [25], the LCD-Array for TBP can also detect unknown strains or species through conserved oligonucleotide “catch all” probes, representing a whole genus or family. Such amplicons hybridizing with “catch all” probes can be subjected to cloning and DNA sequencing to elucidate their origin. Most generic primer, however, are not able to amplify every variant and/or mutant of the species, genus or family of interest. This limits the detection of all available and yet undetected pathogens [26]. The current microarray was optimized for coverage of as many strains possible of its species or genus reported and deposited in the GenBank repository. Furthermore, the reliance of a species-specific and a genus group-specific probe minimizes the likelihood of false negatives at least on genus level. Since “catch all” probes are efficiently hybridizing with complementary amplicons, a depleting effect can occur if the DNA concentration of the respective pathogen is relatively low (Figure 4). Related to the tested concentration, the species-specific probes were able to hybridize in all cases, sometimes with a weaker intensity (Figure 2: *A. sp*. ‘Hadesa’), however with a relatively high copy number. The reason of this discrepancy in comparison to other controls with the same copy number (Figure 2: *T. mutans*) which produce a stronger signal may be optimization issues for the amplification of the *Anaplasma*/*Ehrlichia* template.

In most of the cases the pathogen in the field-collected sample produced a hybridization signal above the cut-off value hence recognized by the software as a positive pathogen identification. Pathogens showing hybridization with a lower than the cut-off value were considered negative, even if in conformity with the previously obtained Sanger sequencing result. Such cases are better understood when used in a larger sample size. Therefore, recurrent appearance on the LCD-Array below the cut-off value of a doubtful pathogen and its distribution can be an indicator of its presence in the area.

In our sample subset, the inconclusive appearance of *E. canis* below cut-off may be due to the degradation of DNA in the original sample. The cattle samples were collected from April 2014 to June 2015, originally preserved in trehalose solution for transportation [27] and stored at −20 °C between analyses.

No cross reactivity among probes and plasmids were observed in the LCD-Array during testing. A number of the negative samples by gel electrophoresis and Sanger sequencing did not show probe hybridization. Some of the negative samples by PCR show hybridization on the array above the valid cut-off threshold. All field samples tested positive by PCR were confirmed by the LCD-Array being infected with TBPs.

One of the most critical aspects in epidemiological surveillance to avoid false positives and negatives relies on the workflow upstream the LCD-Array or sequencing. From the sampling to the DNA/RNA extraction, appropriate management of the samples is mandatory as inaccurate handling may lead to loss of DNA or contamination [28]. Amplification with Uracil instead of Thymine nucleotides and the addition of Uracil N-glycosylase is one approach to prevent carryover amplicon contamination [29]. Whereas the LCD-Array provided one false negative (*E. canis*), no false positives were confirmed. Optimization of calculation of the cut-off value could reduce the error rate further.

The addition of all three PCR products per sample at the same ratio helped the follow up of the sensitivity and possible cross contamination in case of high copy numbers. Tests using different ratios showed *Anaplasma* being the least sensitive followed by *Rickettsia* and *Theileria* having a higher sensitivity (Figure 2). Consequently, pathogens in low concentration may be overlooked, particularly of *Anaplasma*. This could be improved by protocol optimization or by starting the amplification using a higher template volume (2 or 5 µL) increasing the final concentration. Touch-down PCR program prior to hybridization have showed outstanding results in increasing sensitivity and yield which is of great value as long as the specificity is not hampered [30].

## 5. Conclusions

The presence of some of the most important non-viral TBPs for livestock on this LCD-Array, including those with zoonotic potential is a valuable asset. In the future, more groups of TBPs including arboviruses or helminths can be added. Although, the production of microarrays with species coverage of 100 and more is possible, the implementation of a running pipeline for diagnostic analyses is more challenging and herein not addressed. With the novel LCD-Array, a sequencing facility which is often lacking in developing countries is not compulsory. Additionally, post-PCR processing times are as short as 45 min, making immediate reporting and response after TBP outbreaks possible. Low- or non-pathogenic species must be incorporated for subsequent identification. Moreover, the better prospect to find endemic or newly introduced species can contribute to the understanding of possible heterologous reactivity responsible of the host health state.

## Figures and Tables

**Figure 1 tropicalmed-04-00064-f001:**
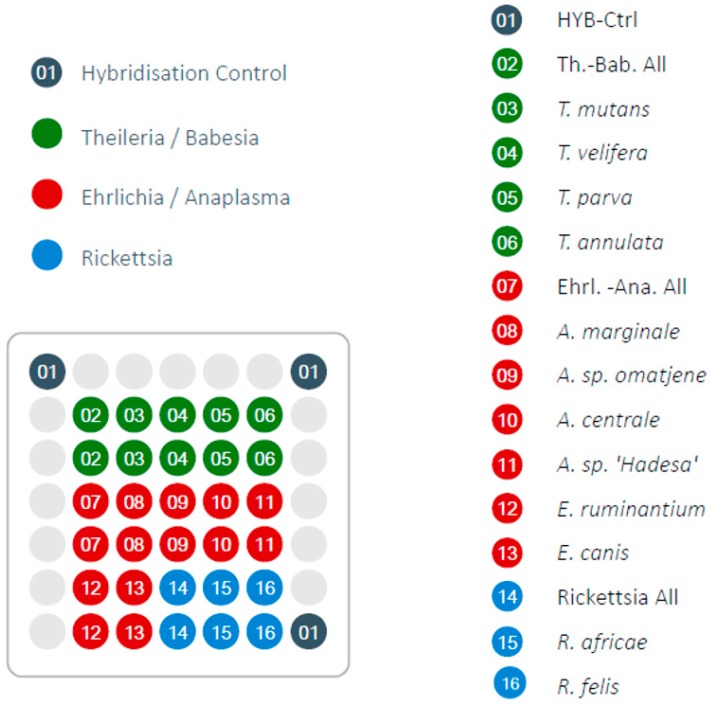
Design of LCD-Array for tick-borne pathogens indicating the screened species and genera. Light grey circles are blank positions.

**Figure 2 tropicalmed-04-00064-f002:**
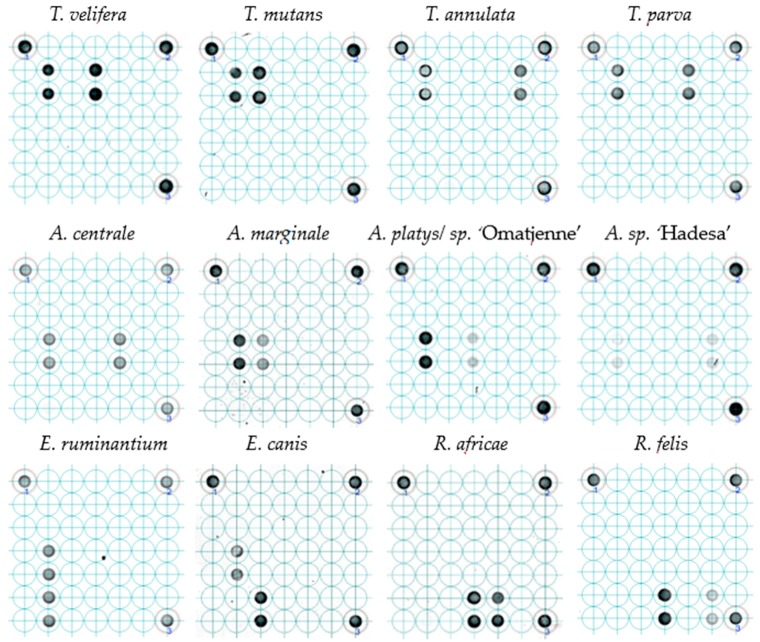
Probe hybridization of LCD-Array of tick-borne pathogens. The dark spots indicate hybridization of plasmids with species-specific inserts to the probe spotted on the array in duplicates. The faint spots indicate lower concentrations in the respective PCR products. The three spots in the corners are internal kit controls. For each of the tested positive controls (plasmids), the concentration came from the first dilution not producing a visible product in agarose gel.

**Figure 3 tropicalmed-04-00064-f003:**
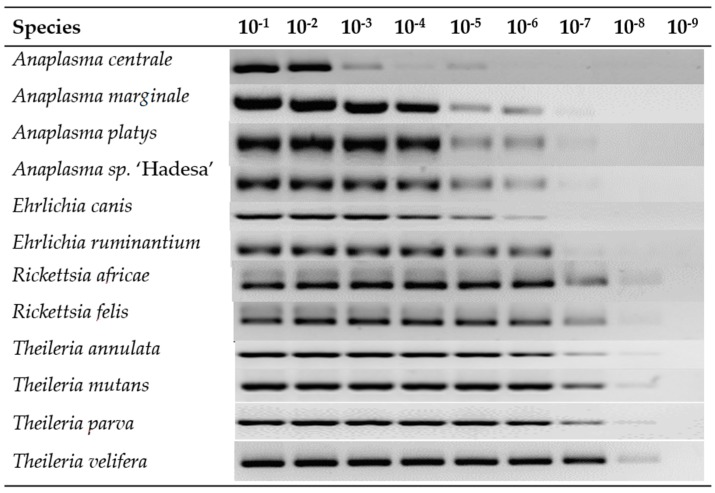
Serial dilution of plasmid amplicons in a 1.5% agarose gel electrophoresis. The last visible band determines the limit of detection which is the lowest dilution detectable on the agarose gel.

**Figure 4 tropicalmed-04-00064-f004:**
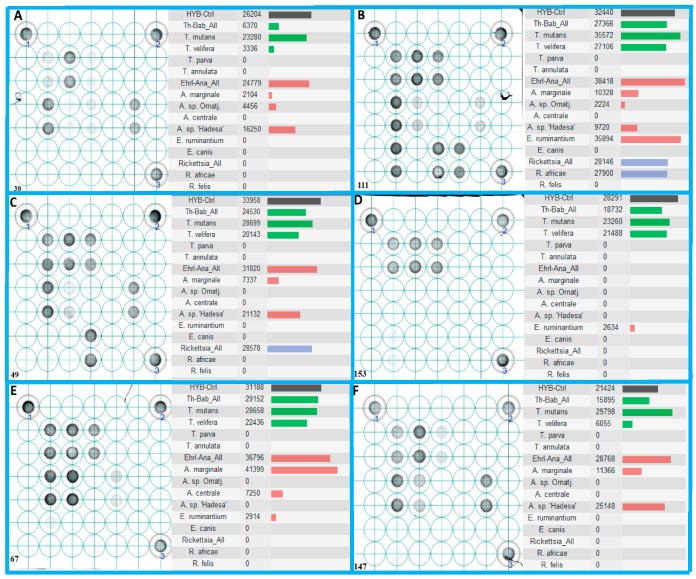
Probe hybridization of six field-collected blood samples (**A**–**F**) on LCD-Array detecting tick-borne pathogens, with 1–3 representing the proprietary kit controls. All shown specimens exhibit co-infections with a minimum of three tick-borne pathogens. The right half of each delimited box shows the hybridization intensity of the corresponding target probe duplicates (Kit control: Black color bar; *Babesia*/*Theileria*: green color bar; *Anaplasma*/*Ehrlichia*: red color bar; *Rickettsia*: blue color bar). Results below the cut off value of 2000 are considered negative.

**Table 1 tropicalmed-04-00064-t001:** Primer pairs used for identification of tick-borne pathogens.

Genus	Gene Target	Primer Sequence	Annealing Temp.	Amplicon Size [bp]	Reference
*Babesia*/*Theileria*	18S rRNA	GAC ACA GGG AGG TAG TGA CAA G	57 °C	460–500	[20]
		b-CTA AGA ATT TCA CCT CTG ACA GT			
*Anaplasma*/*Ehrlichia*	16S rRNA	AGA GTT TGA TCM TGG YTC AGA A	55 °C	460–520	This study
		b-GAG TTT GCC GGG ACT TYT TC			
*Rickettsia*	16S rRNA	GAA CGC TAT CGG TAT GCT TAA CAC A	64 °C	350–400	[10]
		b-CAT CAC TCA CTC GGT ATT GCT GGA			

b- biotin label at 5′ end.

**Table 2 tropicalmed-04-00064-t002:** Limit of detection (LOD) of LCD-Array for tick-borne pathogens measured in the lowest detectable dilution of the PCR product.

Species	Copies/µL Pre-PCR *	LOD Post-PCR *	LOD LCD-Array
*Anaplasma centrale*	75	10^−5^	10^−8^
*Anaplasma marginale*	31	10^−7^	10^−8^
*Anaplasma platys*	28	10^−7^	10^−8^
*Anaplasma sp.* ‘*Hadesa*’	34	10^−7^	10^−8^
*Ehrlichia canis*	60	10^−6^	10^−8^
*Ehrlichia ruminantium*	40	10^−7^	10^−8^
*Rickettsia africae*	3	10^−8^	10^−9^
*Rickettsia felis*	2	10^−8^	10^−9^
*Theileria annulata*	6	10^−8^	10^−9^
*Theileria mutans*	3	10^−8^	10^−9^
*Theileria parva*	7	10^−8^	10^−9^
*Theileria velifera*	1	10^−8^	10^−9^

* Detected on agarose gel electrophoresis.

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
