# Peer review of "Development of a Low-Density DNA Microarray for Detecting Tick-Borne Bacterial and Piroplasmid Pathogens in African Cattle"

_tropicalmed, 2019, doi:10.3390/tropicalmed4020064_

Round 1

Reviewer 1 Report

General comment:

1. As Babesia infections in cattle are common, it seems odd none was not detected in the small sample of cattle from Northern Cameroon.  The authors could comment on this in the Discussion.

2. The Discussion might helpfully consider how the impressive technique could be developed further.  For example, could arboviruses be included in the platform?

3. Suggestions for improvements to the English are listed below.

Specific comments:

Line 21: change ‘towards its’ to ‘for’

Lines 24-25: Qualify the statement regarding limit of detection eg. …depending on species.

Line 27: change ‘less’ to ‘fewer’

Line 30: ‘- also in co-infections to one another’ to ‘, including co-infections of individual animals.’ 

Line 40: change ‘of bacteria’ to ‘as bacteria’

Line 44: change ‘those techniques’ to ‘such techniques’

Line 53: change ‘Its function bases’ to ‘The technology is based on’

Line 62: change ‘the presently mostly’ to ‘the current most commonly’

Line 63: change ‘where…to’ to ‘to which the probes of interest are linked prior to testing’

Line 115: ‘Plasmids’

Line 131: Are Chip-Spin centrifuges likely to be present in veterinary laboratories in endemic countries?  If not, can an alternative methodology be provided?  

Line 141-142: change to ‘Sensitivity was assessed by testing a range of dilutions for each plasmid (Table 2).

Line 151: change to ‘Statistical p-values’   

Fig. 2: Explain that each test is in duplicate. The figure legend refers to ‘dark spots.’  What about the faint spots?  Line 215 correct ‘a plasmids’

Fig. 4:  Change legend to: ‘Probe hybridization of six field samples (A – F) on LCD-Array…’. 

Line 227: change ‘materialized by…’ to ‘expressed as the length of each bar using the color code…’

Line 245: change ‘pose’ to ‘poses’

Line 247: change to ‘is often unsuccessful or distorts..’

Line 254: change to ‘which was 10 to 100 times higher than that obtained by PCR and sequencing’

Line 270: change to ‘primers’

Line 279-281: This sentence does not make sense. Please re-write.

Line 282: change to ‘In most cases, infections in field samples produced hybridisation values above the cut-off and were consequently recognised by the program as positive samples.’

Line 286-288:  This sentence is difficult to understand.  Please re-write.

Line 289: change ‘the non-apparition of’ to ‘the inability to detect’

Line 296-297: This sentence does not make sense. If 100% PCR positives were confirmed as infected with TBPs, why is this ‘In general.’  Please explain the implied exceptions.

Line 318: change to ‘reaction’

Lines 319-321: This sentence is difficult to understand.  Please re-write.

Line 329: what is meant by ‘Experiment can be directly overtaken after sampling’?

Author Response

Dear reviewer,

thank you very much for the very precise, clear and constructive comments. We are thankful for the time you allocated to the review of the manuscript. Each of your comments has been carefully considered and the open points addressed.

Specific comments for improvements can be seen in the revised manuscript and specific responses to raised questions are attached.

The Authors

Reviewer 2 Report

See the attached comments. 

Author Response

Dear reviewer,

we appreciate very much your precise, clear and constructive comments. Hence we are thankful for the thoughtful and thorough review of the manuscript. Each of your comment has been carefully considered and questioning points answered.

Comments for improvements can be seen in the revised manuscript and responses to raised questioning are attached.

 Kind regards,

The Authors

Round 2

Reviewer 2 Report

The revised manuscript has been improved compared to the previous version. However, there area few minor things (as listed below) that needs to be corrected.

The 2.1 should be "Sample origin, DNA extraction, PCR and Sanger sequencing"

- Throughout the "Materials and Methods" section, the authors did not address my comment to mention the manufacturer address ( both city and country).

- It is unclear why Figure 1 is placed both in the main text and in the supplemental file. If the authors would like to put the Fig. 1 in supplement, mention accordingly in the main text. Whichever way you want to go, please be consistent.

-Line 109: (n=31) 

Author Response

Dear Reviewer,

The authors appreciate very much your valuable comments which have greatly improved the manuscript. Each of the comments have been considered and questioning points answered.     Comments for improvements can be seen in the revised manuscriot and responses to raised questioning are attached.

Regards,

The Authors
